# An Artificial Intelligence-Enhanced Blood Pressure Monitor Wristband Based on Piezoelectric Nanogenerator

**DOI:** 10.3390/bios12040234

**Published:** 2022-04-11

**Authors:** Puchuan Tan, Yuan Xi, Shengyu Chao, Dongjie Jiang, Zhuo Liu, Yubo Fan, Zhou Li

**Affiliations:** 1Beijing Advanced Innovation Centre for Biomedical Engineering, Key Laboratory for Biomechanics and Mechanobiology of Ministry of Education, School of Biological Science and Medical Engineering, Beihang University, Beijing 100191, China; tanpuchuan@binn.cas.cn (P.T.); xiyuan@binn.cas.cn (Y.X.); liuzhuo@buaa.edu.cn (Z.L.); 2CAS Center for Excellence in Nanoscience, Beijing Key Laboratory of Micro-Nano Energy and Sensor, Beijing Institute of Nanoenergy and Nanosystems, Chinese Academy of Sciences, Beijing 101400, China; chaoshengyu@binn.cas.cn (S.C.); jiangdongjie@binn.cas.cn (D.J.); 3School of Nanoscience and Technology, University of Chinese Academy of Sciences, Beijing 100049, China

**Keywords:** biosensors, self-powered, piezoelectric nanogenerator, deep learning, artificial intelligence

## Abstract

Hypertensive patients account for about 16% to 37% of the global population, and about 9.4 million people die each year from hypertension and its complications. Blood pressure is an important indicator for diagnosing hypertension. Currently, blood pressure measurement methods are mainly based on mercury sphygmomanometers in hospitals or electronic sphygmomanometers at home. However, people’s blood pressure changes with time, and using only the blood pressure value at the current moment to judge hypertension may cause misdiagnosis. Continuous blood pressure measurement can monitor sudden increases in blood pressure, and can also provide physicians with long-term continuous blood pressure changes as a diagnostic reference. In this article, we design an artificial intelligence-enhanced blood pressure monitoring wristband. The wristband’s sensors are based on piezoelectric nanogenerators, with a high signal-to-noise ratio of 29.7 dB. Through the transformer deep learning model, the wristband can predict blood pressure readings, and the loss value is lower than 4 mmHg. By wearing this blood pressure monitoring wristband, we realized three days of continuous blood pressure monitoring of the subjects. The blood pressure monitoring wristband is lightweight, has profound significance for the prevention and treatment of hypertension, and has wide application prospects in medical, military, aerospace and other fields.

## 1. Introduction

Hypertension refers to a disease with persistently high arterial blood pressure, which often occurs in middle-aged and elderly people, alcoholics and obese people [1]. It is estimated that hypertensive patients in the world account for about 16% to 37% of the global population [2]. Every year, about 9.4 million deaths worldwide are related to high blood pressure, accounting for about 18% of all deaths [3]. Blood pressure measurement is an important method for the initial diagnosis of hypertension. Existing blood pressure measurement methods are divided into two types: invasive and non-invasive. Only critically ill patients and complicated situations require invasive measurement, which involves sending a catheter from a peripheral artery to the aorta through percutaneous puncture and connecting the catheter’s end to a monitoring and pressure measurement device [4]. The traditional non-invasive blood pressure measurement method is the cuff compression method. Medical centers often use mercury sphygmomanometers, and household sphygmomanometers are generally electronic sphygmomanometers [5].

With the popularity of wearable electronics, concepts such as mobile medical and intelligent medical have also been proposed [6,7,8]. Consumers are more inclined to obtain their own health information anytime, anywhere, rather than being limited to medical facilities [9,10,11]. Therefore, this also means higher functional requirements for wearable devices. For hypertensive patients, if real-time blood pressure measurement can be achieved, on the one hand, the occurrence of sudden hypertension complications can be monitored, and the physician can be notified in time for rescue; on the other hand, long-term continuous blood pressure information can be provided for the physician [12].

The cuff pressure method cannot abandon equipment such as pressure pumps and airbags, which is contrary to the miniaturization and high integration of wearable electronic. Therefore, if real-time measurement of blood pressure is to be achieved, new testing and processing methods need to be sought [13]. The pulse wave signal is easy to collect, the signal is strong and it contains a lot of cardiovascular health information of people. Currently, the pulse wave signal has received extensive attention, and various analysis algorithms for pulse wave have sprung up. On the other hand, the vigorous development of artificial intelligence has provided more ideas for digital signal processing of wearable medical devices [14]. Using deep learning of artificial intelligence to analyze massive data can reduce manual intervention, mine data information, and improve the accuracy [15,16].

Another problem with wearable devices is power consumption [17]. Existing sensors used in wearable devices generally require additional excitation voltage to work, which increases the complexity of the system. On the other hand, it also increases power consumption [11,18]. Active sensors, such as mechanics sensors based on triboelectric nanogenerators and piezoelectric nanogenerators (PENG), require no external power supply and have low power consumption [19,20,21]. At the same time, these two sensors also have the advantages of low cost and a wide selection of materials, and flexible materials with good biocompatibility can be selected [22,23,24]. The above advantages make active sensors based on nanogenerators have wider application prospects in biomedical monitoring.

## 2. Materials and Methods

Molds for the blood pressure prediction wristband (BPPW) were designed and printed using a three-dimensional printer (Raize 3D) and polylactic acid (PLA) printing supplies. For wireless data acquisition and transmission, a commercial Bluetooth board was used. A linear motor (LinMot E1100, Suzhou, China) was utilized to continuously impart periodic mechanical traction to the BPPW to maintain the operating cycle. A Keithley 6517 electrometer (Beijing, China) was used to measure the open-circuit voltage, short-circuit current and short-circuit charge of the BPPW, and the data were obtained and recorded using an oscilloscope (LeCroy HDO6104, New York, NY, USA). A wireless motion monitoring system based on a BMD101 board was used to record data from the BPPW. A commercial physiological recording analyzer system (PRAS), BIOPAC: MP150, was used for a pulse acquisition test. Additionally, the blood pressure value is obtained by an electronic blood pressure monitor: yuwell YE655A (Shanghai, China).

## 3. Results

An artificial intelligence-enhanced blood pressure predict wristband (BPPW) was developed in this article, which integrates a sensor based on PENGs. Using this sensor, the wristband can obtain the wearer’s pulse wave signal, and then through compared with the established artificial intelligence model, this wristband can realize the prediction of the wearer’s blood pressure. The wristband consists of five parts, including a rubber strap, a PLA shell, a lithium-ion battery, a Bluetooth module and a PENG based sensor (Figure 1A reveals the structure and materials of the BPPW, respectively). Figure 1B shows the wearer wearing a BPPW. Figure 1C shows the BPPW alone. The overall length of the BPPW is 26 cm, the strap part is 2 cm wide and 2 mm thick, and the PLA shell is 4.8 cm long, 2.8 cm wide and 2 cm high.

Through the BPPW, we can implement continuous blood pressure monitoring for the wearer. The whole monitoring system consists of three parts, including the device side, the processing side and the application side (Figure 1D). The device side is the main part of the wristband. The main work of this part is to collect pulse information on the spot and transmit the signal to the processing side through the Bluetooth module. Once the processing side receives these data, after a series of data processing, the collected data will be compared with the previously trained deep learning model, and the prediction result of blood pressure is obtained. It is conceivable that after the completion of the industrialization of the BPPW, for the long-term blood pressure data of the wearer, the system can upload the long-term blood pressure of the wearer to the cloud, or notify the sudden high blood pressure or low blood pressure to physician. In the current study, due to the limitations of experimental conditions, we adopt the method of off-site calculation, upload the data to the processing end and then perform the calculation. After commercialization in the future, we can consider the local calculation method and directly merge the device side and processing side.

The sensor part of BPPW is based on PENG, which is divided into four layers from top to bottom, which are the package layer (upper layer), the generation layer, the structure layer and the package layer (lower layer). The material used in the package layer is polytetrafluoroethylene (PTFE), the material used in the power generation layer is polyvinylidene fluoride (PVDF) film, and the structural layer is 3D printed PLA with microstructures (Figure 2A). The microstructure on the structural layer is helpful for the output enhancement of the generator. A contraction and relaxation of the heart constitutes a cycle of mechanical activity called the cardiac cycle. The cardiac cycle generates pulses that travel along the arteries, transporting blood throughout the body (Figure 2B). We set the sensor on the wrist, and the pulse signal can be collected through the radial artery [25]. The motion of one cardiac cycle acts on the sensor, corresponding to one power generation cycle of the generator. As shown in Figure 2C, one power generation cycle of the generator can be divided into four steps. The first is the initial state, where the sensor is in equilibrium and no current is produced. Next, the sensor is subjected to the action of external force, the PVDF is bent, the induced charge is generated on both sides of the film, and the external circuit has current passing through. Then, the external force reaches the peak, the bending degree of PVDF reaches the maximum, and the induced charge on both sides also reaches the maximum. When the external force is removed, the PVDF recovers from the bending state, the induced charge on both sides of the film decreases, and the external circuit has a reverse current flow. Finally, it returns to the initial state and enters the next cycle. Figure 2D shows the situation when the BPPW we designed and a commercial pulse sensor monitor the same subject at the same time. The output results of the BPPW and the commercial pulse sensor are relatively close, indicating that our sensor can reflect the real situation of the pulse signal. Figure 2E–G, respectively, show the open-circuit voltage, short-circuit current and charge transfer of the sensor generated by blood pressure directly. The peak voltage output of the BPPW is 0.41 V, the peak current output is 0.21 μA and the single charge transfer amount is about 45 nC.

In order to obtain a sensor with higher output performance and higher signal-to-noise ratio facing the blood pressure measurement, we explored the microstructure of the sensor. We set up four kinds of sensors, the structure layer structure of these four sensors is different, as shown in Figure 3A. The printing material selected is hot-melt PLA, the temperature of the 3D printer’s nozzle is set to 215 degrees Celsius, and the temperature of the baseplate is set to 60 degrees Celsius. We drew the model of the structural layer in advance, sliced the model and uploaded it to the 3D printer for printing. The printing time was about 42 min. After waiting for the model to cool, peel the sensor from the baseplate. The model is then treated with surface polishing liquid, placed in a fume hood and after the polishing liquid is air-dried, the processed structural layer model is obtained. These four types are no microstructure type, cylindrical type, prismatic type and no structure layer type. Among the four structures, the cylindrical and prismatic types with microstructures have the higher output, and the cylindrical type has a significant performance improvement compared to the prismatic type. Sensors without structured layers also have an output, but the output is not as high. Furthermore, the sensor without microstructure has no output because the generation layer is limited by the structure layer. Next, we explored the height of the microstructure. Experiments show that with the increase in the height of the microstructure, the output of the sensor increases, but when the height of the microstructure exceeds 2 mm, the output decreases instead (Figure 3B). Finally, we explore the cylinder size and spacing, and Figure 3C shows that the sensor output decreases with increasing cylinder size and spacing. Based on the above experiments, we finally choose a cylindrical structure layer, the height is set to 2 mm and the spacing is set to 1 mm. Under this condition, the signal-to-noise ratio of the sensor is 29.7 dB. After the sensor generates current, it needs to be processed by some noise reduction circuits. Figure 3D shows the circuit diagram of the BPPW. The circuit of the whole BPPW consists of three parts, the sensor part, the Bluetooth module part and the computer part. After the TENG converts the mechanical signal into an electrical signal, a noise reduction circuit consisting of a capacitor and a resistor needs to be connected to its positive electrode. After the signal is transmitted to the Bluetooth module, the data is transmitted to the computer in real time through wireless transmission for processing and analysis. The Bluetooth module selected is the BMD101.

The difference of sensor output mainly comes from the different deformations of PVDF film. Under the condition of consistent force, the deformation of the sensor comes from the pressure exerted by the structural layer on the power generation layer. The greater the pressure, the greater the deformation of the PVDF film. Therefore, the cylinder microstructure with the smallest contact area exerts the greatest pressure, deforms the PVDF film the most and generates the greatest voltage output. For the sensor without a structure layer and the sensor without a microstructure, since the contact area between the structure layer and the power generation layer of the sensor without a microstructure is very large, the force exerted by the wrist skin on the sensor can hardly affect the power generation layer of the sensor. So, there is almost no output. The sensor without the structural layer can receive part of the force from the wrist, so it has a weak output. The potential generation result from four microstructure of finite element analysis are also consistent with the results we discussed (Figure 3E–H).

As a sensor that needs to work for a long time, its durability is particularly important. We have performed robustness tests on the sensor by a linear motor more than 10,000 times cycles. As shown in Figure 4A, the sensor is relatively stable before and after the entire fatigue test. Its voltage output is basically stable at about 2.5 V, and the waveform does not change much. Blood pressure fluctuates throughout the day. Generally speaking, the waveform of blood pressure of a human in a day is in the shape of a “spoon”, which is highest in the morning, decreases in the afternoon, and reaches the lowest value after dinner. We performed a continuous blood pressure tracking test on a subject. The left side of Figure 4B shows the subject’s pulse signal gathered by our sensor, and the right side shows the subject’s blood pressure value collected by the electronic sphygmomanometer. Through the pulse data on the left, we found that when the blood pressure of the subjects was relatively high, the peak value of the pulse output signal had little effect, but the small peaks between the main peaks oscillated more violently. Blood travels from the heart to the extremities, and the timing of the pulse waves to each part of the body varies slightly. We set up two sensors and placed them upstream and downstream of the radial artery, with a difference of 5 mm in their center positions (Figure 4C). It can be observed that the signal peak of the sensor at the proximal end always appears before the sensor at the distal end, and the time phase difference between the two signals is about 0.047 s.

We performed 120 pulse signal acquisitions and recorded the subjects’ blood pressure measurements at each acquisition. We intercepted 10 s of data in each sample, and finally combined these 120 sets of data with the corresponding blood pressure labels to build a deep learning regression model. After the original data is subjected to one-dimensional convolution processing (convolution kernel size is 3, stride size is 1, noise reduction processing), it is divided into several batches and placed in the model of the self-supervision–incentive mechanism, then the model is continuously updated with reconstructed parameters, obtains a linear parameter layer and finally performs regression with the blood pressure value through this linear layer (Figure 5A). Since there is no obvious relationship between high and low blood pressure, we did not regress to two blood pressure values once, but established two models to regress to systolic and diastolic blood pressure values, respectively. Figure 5B shows the sample distribution of our database, a total of 120 samples, including 82 males and 38 females. In terms of body mass index (BMI), 26 were underweight (BMI < 20), 62 were normal weight (20 < BMI < 25) and 32 were overweight (BMI > 25). According to the blood pressure measurement values, 26 patients had lower diastolic blood pressure values (value < 65), 78 patients had normal diastolic blood pressure values (65 < value < 85) and 16 patients had higher diastolic blood pressure values (value > 65). There were 26 patients with lower systolic blood pressure value (SBPV) (value < 95), 26 patients with normal systolic blood pressure value (95 < value < 135) and 26 patients with higher systolic blood pressure value (value > 135). Figure 5C shows that the loss value of the model decreases with the number of iterations. It can be seen that the diastolic blood pressure value tends to converge after 48 cycles, the loss value after convergence is about 4 mmHg and the systolic blood pressure value after 56 iterations then tends to converge; the loss value after convergence is about 6 mmHg. Finally, we tested a subject at risk of hypertension with BPPW for three consecutive days (Figure 5D). Based on the collected pulse signals, our model gave a predicted value of blood pressure, where at 12:00 and 16:00 on the first day and at 20:00 on the third day, the subjects’ diastolic blood pressure values all exceeded the normal blood pressure range. This is also consistent with the actual situation.

## 4. Discussion

Hypertension refers to a chronic disease in which blood pressure remains at high levels. If not treated properly, people with high blood pressure are at risk of stroke, coronary heart disease, heart failure, chronic kidney disease and even premature death [2]. At present, the diagnosis of hypertension in hospitals is usually determined by a blood pressure measurement. Considering that people’s blood pressure is variable and some people have white coat hypertension, this diagnosis method has the possibility of misdiagnosis. BPPW is designed to monitor blood pressure continuously and for a long time. By optimizing and redesigning the structure and size of BPPW, we realize the miniaturization and integration of BPPW, which are two important requirements of wearable electronics [26,27,28,29]. The size of BPPW’s shell is within 4.8 cm long, 2.8 cm wide and 2 cm high, which further improves the wearability of BPPW. Another focus of our research is to improve the accuracy of BPPW in blood pressure prediction [11]. The original data collected by BPPW is the pulse wave. The pulse wave can directly reflect the heart rate, but the relationship between pulse wave and blood pressure is not reflected directly. We use a transformer deep learning model to mine the relationship between pulse wave and blood pressure, and reduce the error to 4 mmHg. We tried a variety of deep learning models and found the most suitable algorithm. Perhaps, with the further development of artificial intelligence, this prediction result will be further improved [30]. In some applications combining self-powered sensors with artificial intelligence [31,32,33], researchers use traditional machine learning methods to achieve model establishment. The machine learning algorithms used include linear regression, local weighted regression, ridge regression, lasso regression, classification and regression tree, and so on. In our early experiments, we also tried to use these traditional machine learning algorithms. Traditional machine learning methods need to rely on effective feature engineering. Perhaps the important features of blood pressure reflected by the pulse wave are really not easily extracted. We extracted a variety of features to build a regression model, and the results were not satisfactory, with an average error of more than 11 mmHg. Finally, we chose deep learning. Deep learning does not require manual extraction of features [34], but is performed by computers, and the final experimental results obtained by BPPW are ideal. By analogy, when we solve other biosensors problems, if the feature engineering of the original data cannot well reflect the target results, we can choose appropriate deep learning algorithms to help us perform data mining and reduce manual intervention errors caused.

## 5. Conclusions

In this paper, based on the PENG, a mechanical sensor with excellent performance is designed, and its signal-to-noise ratio can reach 29.7 dB. By exploring the influence of different microstructures on the output, it is found that the microstructure of the columnar array can effectively improve the output performance of the sensor. The BPPW is based on a supervised and self-attention deep learning model, which can predict blood pressure by combining the collected pulse wave data with a pre-established regression model. In this work, we compare multiple artificial intelligence algorithms, and finally choose Transformer, whose prediction error is less than 4 mmHg. The BPPW is ingeniously designed, easy to wear and can monitor blood pressure for a long time. It has great application value for the treatment and prevention of hypertension in patients.

## Figures and Tables

**Figure 1 biosensors-12-00234-f001:**
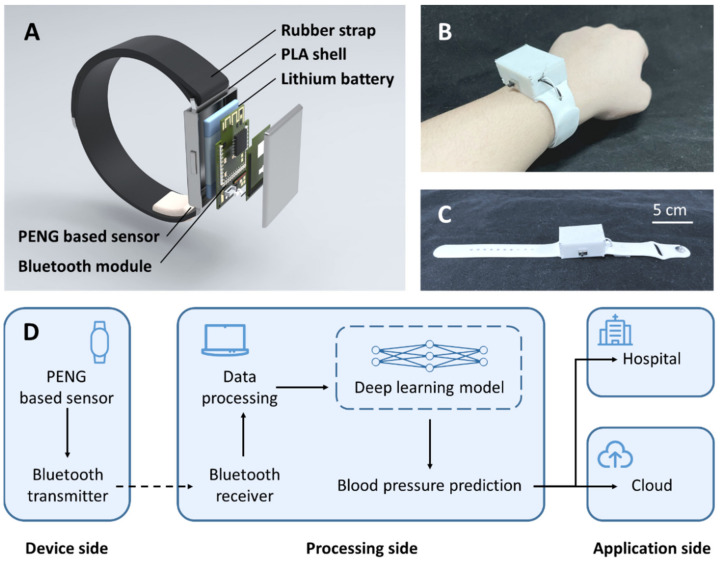
The overview of BPPW. (**A**) The structure of BPPW and the materials used in each part. (**B**) The photograph of subjects wearing BPPW. (**C**) The photograph of BPPW; the whole length of BPPW is 26 cm. (**D**) The design concept of BPPW.

**Figure 2 biosensors-12-00234-f002:**
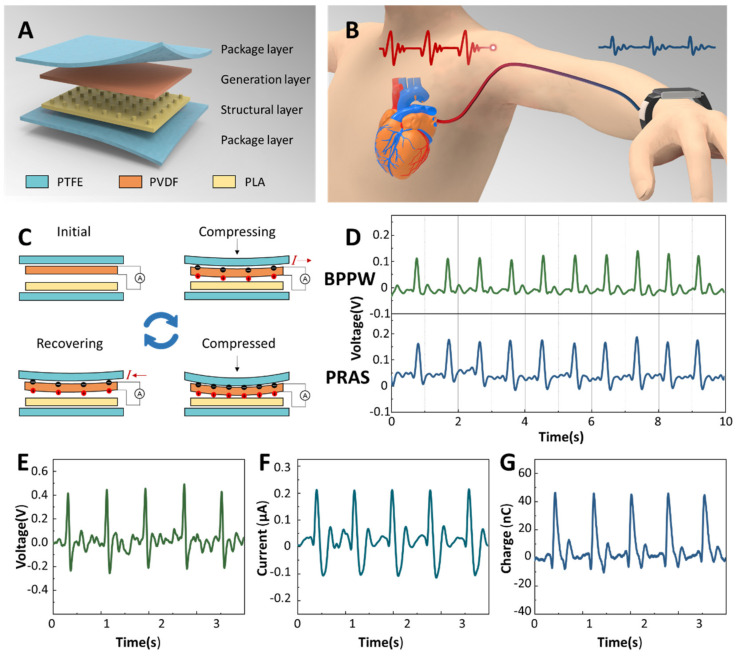
The working principle of BPPW. (**A**) The structure and materials used of sensor in BPPW. (**B**) The pulse wave travels from the heart to the wrist. (**C**) The generation principle of the PENG based sensor used in BPPW. (**D**) Comparison between BPPW and PRAS, the upper is the BPPW signal and the bottom is the PRAS signal. The open-circuit voltage (**E**), short-circuit current (**F**) and the charge (**G**) output of the BPPW.

**Figure 3 biosensors-12-00234-f003:**
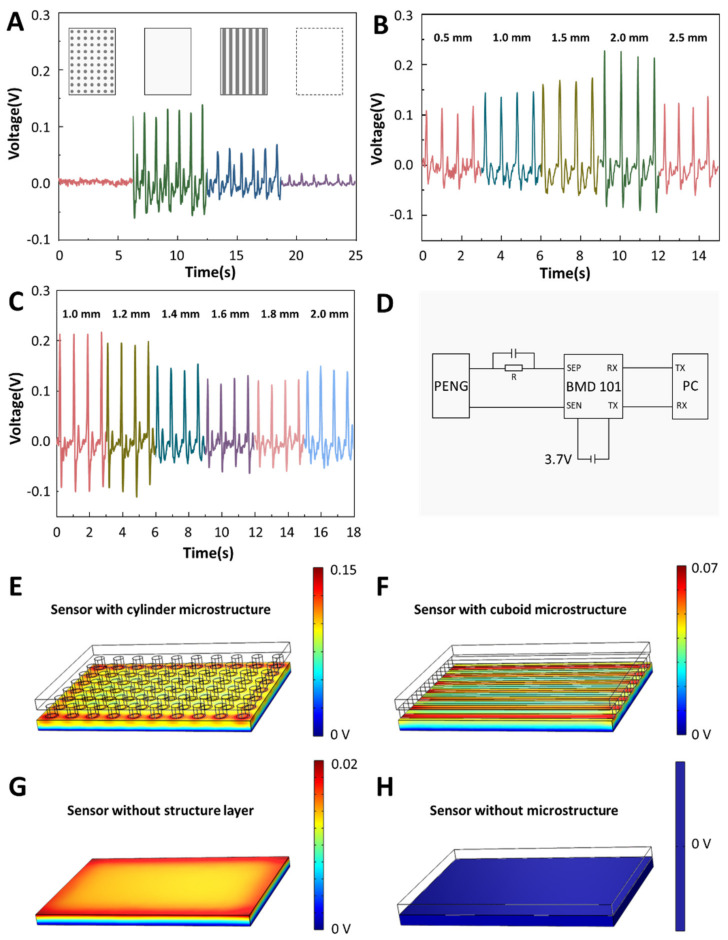
The influence of microstructure differences of structural layers on sensor output. (**A**) The influence of the four structures type on the output, these four structure types including no microstructure type, cylindrical type, prismatic type and no structure layer type. (**B**) The influence of the microstructure length on the output. (**C**) The influence of microstructure spacing on the output. (**D**) The circuit diagram of the BPPW. (**E**–**H**) The finite element analysis of potential generation for four microstructures.

**Figure 4 biosensors-12-00234-f004:**
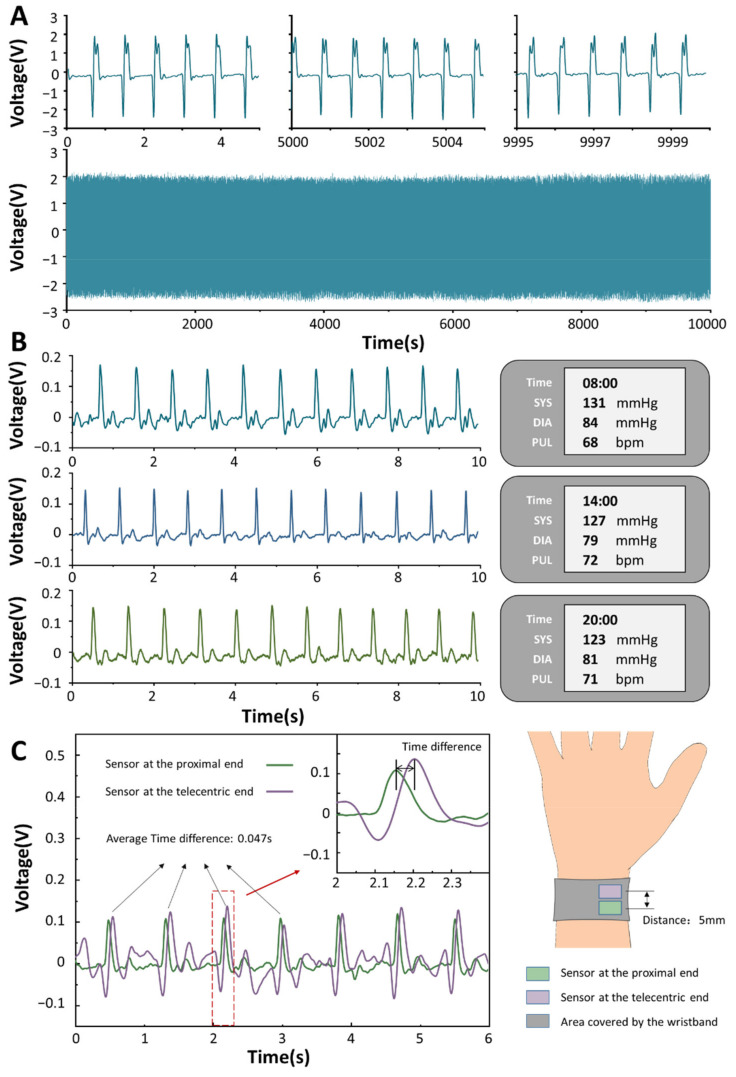
The output performance of the BPPW. (**A**) Robustness test results of BPPW. (**B**) BPPW monitors the subjects’ blood pressure and the readings of the sphygmomanometer at different times of the day. (**C**) The difference in time between two sensors at different locations.

**Figure 5 biosensors-12-00234-f005:**
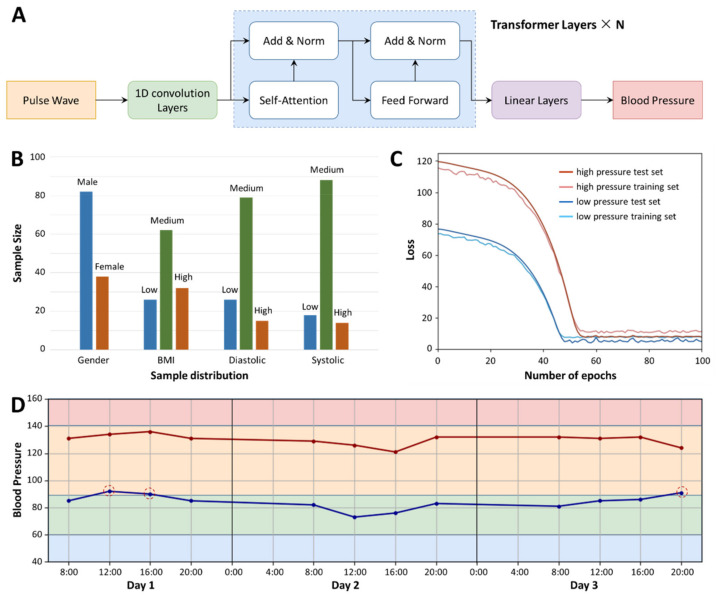
The application of BPPW. (**A**) The process of BPPW’s deep learning model establishment. (**B**) The sample distribution of the training model. (**C**) The loss value decreases with the increase in training epochs. (**D**) A potential hypertensive patient wears a BPPW for three consecutive days, and BPPW predicts his blood pressure. According to BPPW’s prediction, at 12:00 and 16:00 on the first day and at 20:00 on the third day, the subjects’ diastolic blood pressure values all exceeded the normal blood pressure range.

## Data Availability

Not applicable.

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
