# Peer review of "An Artificial Intelligence-Enhanced Blood Pressure Monitor Wristband Based on Piezoelectric Nanogenerator"

_biosensors, 2022, doi:10.3390/bios12040234_

Round 1

Reviewer 1 Report

The manuscript introduces an artificial intelligence-enhanced blood pressure predict wristband (BPPW) which integrates a sensor based on PENGs. This research includes the preparation of sensor, sensor performance test, establishment and analysis of the deep learning regression model. Experimental results show that through the transformer deep learning model, the wristband can predict blood pressure, and the loss value is lower than 4 mmHg. However the following issues should be addressed before it could be published. Detailed Comments: 1. The format of figures’ annotations in the text is different from that on the figures. 2. Please describe in detail the preparation of the microstructures in Figure 2A. 3. What is the scale of the microstructure in Figure 3A?

Reviewer 2 Report

The ms from Tan et al reports a work on the development of an artificial intelligence-enhanced blood pressure monitor wristband sensor based on piezoelectric nanogenerator. Overall, the ms is interesting, with good quality of figure and discussion supported by a good set of experimental data. However,- there is anything about biosensor in this contribution with the blood pressure to be the only "bio" parameter being investigated. The active element of the blood pressure sensor is anything but not bio so my recommendation is to transfer the ms to another journal such as Sensors or Micromachines. Another suggestion is to improve the experimental section which requires more details. As it stands it is very dry.

Reviewer 3 Report

The manuscript “An artificial intelligence-enhanced blood pressure monitor wristband based on piezoelectric nanogenerator” developed PENG based blood pressure monitor wristband using a deep learning model.  The output performance enhanced using microstructured layer, and signal to noise ratio reach 29.7 dB. It has sufficient novelty for publication in Biosensors. However, there also are several issues that the authors need to address carefully first

  1. It is not well indicated whether the output voltages of the PENG presented in this manuscript are measured by external pressure or blood pressure. Clear indications are required to avoid confusion.
  2. In page 4, line 143, author mentioned “the peak output voltage of the BPPW is 3.6 V”, however, it does not match with figure 2E.
  3. Why is the output voltage sometimes in DC form and sometimes in AC form? For example, the output voltage in Figure 2E is perfectly DC, while the output voltages in Figure 3 and Figure 4 are a mixture of DC/AC.
  4. The author suggests that the microstructure of the structural layer enhanced the output performance of the PENG. However, there is no description of the mechanism to improve the output performance of the PENG.
  5. Figure 3D shows the circuit diagram of the BPPW. However, there is no description of the circuit. A more detailed description of the circuit should be included in the manuscript.

Round 2

Reviewer 2 Report

The authors have revised the ms. I'm not against this ms per se but I still believe that Sensors would be better to host this contribution as there are not "bio" elements in this sensor. I mark "accept" but it should be transferred to another more suitable journal.

Reviewer 3 Report

Accept in present form